# Don't Drop Dropout: Optimizing Layer Sparsity for Efficient LLM Training and Inference

**Mostafa Elhoushi** [1]  **Alex Pretko** [2]  **Nolan Dey** [1]  **Bin Claire Zhang** [1]  **Gavia Gray** [1]  **Gurpreet Gosal** [1]
**Abdulrahman Mahmoud** [2]  **Shane Bergsma** [1]  **Joel Hestness** [1]

## Abstract

Layer dropout (a.k.a. stochastic depth) has been shown to enable faster training, higher accuracy, and robustness to zero-shot layer pruning in both language and vision transformers. However, as models and datasets have scaled, dropout—particularly layer dropout—has largely disappeared from LLM pre-training recipes. While some prior work has reported that dropout can degrade accuracy, no comprehensive study has quantified, let alone mitigated, this effect. In this study, we show that layer dropout *should* be used in state-of-the-art LLM training, establishing best practices and scaling analysis for both training and post-training benefits. Concretely, with optimal layer distribution, time schedule, and optimizer hyperparameters, All pre-training experiments were run on Cerebras CS-3 systems.

## 1. Introduction

Pretraining large language models (LLMs) demands extraordinary computational resources (Narayanan et al., 2021), where small improvements in time-to-accuracy can save millions of dollars (Coleman et al., 2019) and reduce carbon emissions (Acun et al., 2023). Historically, regularization techniques improved validation accuracy for given training budgets by reducing overfitting and stabilizing optimization (Moradi et al., 2020; Wang & Manning, 2013; Murugan & Durairaj, 2017). *Dropout* was widely adopted in convolutional networks (Hinton et al., 2012) and early transformers (Vaswani et al., 2017). However, as LLMs scaled to billions of parameters and trillions of tokens, dropout has been largely abandoned (Raschka, 2025). Models trained for a single epoch over massive datasets have little opportunity

for classical overfitting, and empirical evidence suggests activation dropout degrades performance under these conditions (Liu et al., 2025).

One type of dropout, Layer Dropout, also known as Stochastic Depth (Huang et al., 2016), can provide benefits beyond regularization. Unlike activation dropout or unstructured sparsity, which typically do not translate into wall-clock speedups due to sparse-kernel overheads, skipping entire transformer blocks yields structured sparsity that can reduce active training FLOPs almost linearly with the dropout rate (Zhang & He, 2020; Elkerdawy et al., 2021). It also encourages robustness to reduced-depth execution at inference time, enabling a single pretrained model to dynamically adapt to different latency and compute budgets without retraining. This supports zero-shot depth-wise optimizations such as elastic depth (Fan et al., 2020), early exit (Elhoushi et al., 2024), and intermediate layer skipping (Huang et al., 2016; Cai et al., 2021).

Despite layer dropout's promise, its role in state-of-the-art LLM pretraining has never been established through a comprehensive evaluation at scale. Existing evidence is fragmented across model families, dataset sizes, and implementation conventions. Many reported degradations may reflect suboptimal schedules or hyperparameters, rather than fundamental limitations. This leaves a basic unresolved question: *should layer dropout be used in modern large-scale LLM training, and if so, how should it be configured to preserve accuracy while delivering training and deployment benefits?*

We provide the first unified experimental study of layer dropout in LLMs, systematically varying (i) optimizer hyperparameters, (ii) depth-wise distribution and granularity of layer sparsity, and (iii) temporal dropout schedules, across fixed architecture and data. Across hundreds of training runs, we identify configurations that reliably improve training and inference efficiency. Our contributions are:

1. **Improved Compute–Accuracy Trade-offs:** Properly configured layer dropout reduces training FLOPs while achieving validation loss competitive with, and in several cases superior to, dense baselines at scale.

[1]Cerebras Systems [2]MBZUAI. Correspondence to: Mostafa Elhoushi <m.elhoushi@ieee.org>, Joel Hestness <joel@cerebras.net>.

*Proceedings of the $43^{rd}$ International Conference on Machine Learning*, Seoul, South Korea. PMLR 306, 2026. Copyright 2026 by the author(s).

2. **Joint Optimization Framework:** We identify key interactions between dropout configurations, schedules, and optimizer hyperparameters that mitigate degradations observed in prior work.

3. **Depth-Elastic Inference:** We show that the *average training dropout rate* predicts zero-shot robustness to early exit and layer skipping without retraining.

4. **Scaling Analysis and Best Practices:** We analyze performance across model and data scales, recommending a progressively increasing distribution across *depth* paired with a decreasing schedule across *steps*, yielding up to 20% training FLOPs savings and up to 1.4× inference speedup.

## 2. Related Work

**Dropout in large-scale LLM pretraining.** As language models scaled to billions of parameters and trillion-token datasets, explicit regularization techniques—including activation dropout—have largely disappeared from state-of-the-art pretraining recipes. Early decoder-only models such as GPT-3 (Brown et al., 2020) and OPT (Zhang et al., 2022) retained the dropout settings inherited from Vaswani et al. (2017), while later models such as PaLM (Chowdhery et al., 2022) applied dropout only during finetuning, and LLaMA-style models no longer explicitly document its use. Recent empirical studies further suggest that activation dropout can degrade performance in single-epoch, large-data regimes (Liu et al., 2025), reinforcing the prevailing view that dropout is unnecessary or harmful at scale. At the same time, dropout has been shown to remain beneficial in multi-epoch or data-limited settings (Xue et al., 2023), indicating that its utility is highly regime-dependent.

Importantly, techniques originally introduced as regularizers may persist in modern LLM training for reasons unrelated to overfitting prevention. Weight decay, for example, has been shown to primarily influence optimization dynamics rather than classical generalization in large-scale pretraining (D'Angelo et al., 2024). This motivates re-examining dropout—particularly structured variants—without assuming that its value must stem from regularization in the traditional sense.

**Layer dropout across scale.** Layer dropout was originally proposed to stabilize optimization in very deep residual networks (Huang et al., 2016) and has since become standard in large-scale vision models. However, its optimal strength has been observed to diminish as dataset scale increases: for example, ConvNeXt models trained on ImageNet-22K require substantially lower dropout rates than those trained on ImageNet-1K (Liu et al., 2022). A similar pattern appears in language modeling. Progressive Layer Dropout (Zhang & He, 2020) and LayerDrop (Fan et al., 2020) reported improved convergence and robustness in BERT-era, multi-epoch settings on relatively small corpora. In contrast, more recent work applying layer dropout to decoder-only LLMs trained on large token budgets has reported non-negligible accuracy degradation (Elhoushi et al., 2024), suggesting that naive extensions of earlier recipes may not transfer to modern regimes. To date, the literature lacks a controlled, large-scale evaluation that reconciles these conflicting findings by systematically varying dropout configurations and optimizer settings.

**Training-aware approaches to depth elasticity.** Layer dropout relates to training-aware methods for inference-time efficiency. In compression, quantization-aware training (QAT) improves post-training quantization (PTQ) by exposing models to reduced precision during optimization (Stock et al., 2021). Similarly, depth-aware training makes models robust to reduced depth at inference. Prior work achieves depth elasticity via auxiliary losses, routers, or adapters, including early-exit models (Jamialahmadi et al., 2025), routing-based skipping (Jiang et al., 2024; Raposo et al., 2024), and elastic architectures like Once-for-All (Cai et al., 2020), MatFormer (Devvrit et al., 2024), and Nemotron-Elastic (Taghibakhshi et al., 2025).

Layer dropout occupies a distinct position within this landscape: it induces robustness to depth-wise inference optimizations directly during pretraining, without modifying the model architecture or introducing additional losses. Prior work demonstrated that this can enable elastic inference at small scales (Fan et al., 2020), but whether similar benefits can be realized at modern LLM scales without sacrificing base-model accuracy has remained unresolved.

## 3. Methodology

In our experiments, we train decoder-only transformers following the architecture of Celerity models (Bergsma et al., 2025b): ALiBi position embeddings (Press et al., 2022), squared ReLU activations (Zhang et al., 2024b), and Llama3 vocabulary (Grattafiori et al., 2024). Specific architectural dimensions for all model sizes are detailed in the Appendix. Our datasets are obtained from a diverse corpus of natural language text and code.

To develop best practices and quantify the effects of layer dropout, we first identify optimal hyperparameters for each dropout rate (Sec. 5). We then determine optimal granularity (Sec. 6) and configuration (Sec. 7). Following Hoffmann et al. (2022), these experiments utilize a compute-optimal budget of 20 tokens-per-parameter (TPP) at each model size. Subsequently, we evaluate benefits across various depth-wise inference optimizations (Sec. 8), then quantify accuracy as training scales to larger datasets (Sec. 9).

## 4. Preliminary

**General Formulation of Layer Dropout**  We start by denoting residual layer $\ell \in \{0, \ldots, L-1\}$, of an $L$ layer neural network at training step $t \in \{0, \ldots, T-1\}$, as:

$$\mathbf{H}^{\ell+1,t} = \mathbf{H}^{\ell,t} + f^\ell(\mathbf{H}^{\ell,t}) \qquad (1)$$

where, in the domain of natural language processing, activation tensor $\mathbf{H} \in \mathbb{R}^{B \times S \times d}$, $B$ is batch size, $S$ is sequence length, $d$ is hidden dimension.

When layer dropout is applied with rate $p^{l,t}$, the operation of the layer during training at step $t$ becomes:

$$\mathbf{H}^{\ell+1,t} = \mathbf{H}^{\ell,t} + r_{\text{train}}^{\ell,t} \mathbf{M}^{\ell,t} f^l(\mathbf{H}^{\ell,t}) \qquad (2)$$

where mask $\mathbf{M}^{\ell,t} \in \{0,1\}^B \sim \text{Bernoulli}(1 - p^{\ell,t})$ is a Bernoulli random vector, and $r_{\text{train}}$ is a scaling factor applied during training. $r_{\text{train}}$ is defined differently in different layer dropout literature, and we will discuss our choice later.

The $b^{\text{th}}$ sequence of $\mathbf{H}$ during training is now equal to[1]:

$$\mathbf{H}^{\ell+1,t}[b,:,:] = \begin{cases} \mathbf{H}^{\ell,t}[b,:,:], \\ \qquad \text{with probability } p, \\ \mathbf{H}^{\ell,t}[b,:,:] + r_{\text{train}}^{\ell,t} f^l(\mathbf{H}^{\ell,t}[b,:,:]), \\ \qquad \text{with probability } 1 - p. \end{cases} \qquad (3)$$

While layer dropout could be implemented during training by executing $f(\mathbf{H}^{\ell,t})$ on all sequences $b \in \{0, 1, ..., B-1\}$ of $\mathbf{H}$, and multiplying its output by $\mathbf{M}^{\ell,t}$, a more efficient implementation would be to only execute $f(\mathbf{H}^{\ell,t})$ on sequences $b \in \{b_i \mid \mathbf{M}^{\ell,t}[b_i] = 1\}$. This leads to a saving of a portion $p$ of training FLOPs of the layer.

During inference, dropout is typically disabled and a distinct scaling factor, $r_{\text{eval}}^\ell$, is applied:

$$\mathbf{H}^{\ell+1} = \mathbf{H}^\ell + r_{\text{eval}}^\ell f^\ell(\mathbf{H}^\ell) \qquad (4)$$

**Layer Dropout for a Transformer**  We denote the operation of layer $\ell \in \{0, \ldots, L-1\}$ of an $L$, layer transformer model, at time step $t \in \{0, \ldots, T-1\}$, during training as:

$$\begin{aligned} \mathbf{Z}^{\ell,t} &= \mathbf{X}^{\ell,t} + f_{\text{attn}}^l(\mathbf{X}^{\ell,t}) \\ \mathbf{X}^{\ell+1,t} &= \mathbf{Z}^{\ell,t} + f_{\text{ffn}}^l(\mathbf{Z}^{\ell,t}) \end{aligned} \qquad (5)$$

where $\mathbf{X}, \mathbf{Z} \in \mathbb{R}^{B \times S \times d}$, $f_{\text{attn}}^\ell$ is the attention layer and $f_{\text{ffn}}^\ell$ is the feed-forward network (FFN).

When layer dropout is applied with rate $p^{\ell,t}$, the operation at transformer layer, $\ell$, step, $t$, during training becomes:

$$\begin{aligned} \mathbf{Z}^{\ell,t} &= \mathbf{X}^{\ell,t} + r_{\text{train}}^{\ell,t} \mathbf{M}_{\text{attn}}^{\ell,t} f_{\text{attn}}^\ell(\mathbf{X}^{\ell,t}) \\ \mathbf{X}^{\ell+1,t} &= \mathbf{Z}^{\ell,t} + r_{\text{train}}^{\ell,t} \mathbf{M}_{\text{ffn}}^{\ell,t} f_{\text{ffn}}^\ell(\mathbf{Z}^{\ell,t}) \end{aligned} \qquad (6)$$

---

[1]For neuron dropout, i.e., the default variant of dropout introduced by (Hinton et al., 2012), $\mathbf{M} \in \{0,1\}^{\{B \times S \times d\}}$.

and during inference becomes:

$$\begin{aligned} \mathbf{Z}^{\ell,t} &= \mathbf{X}^{\ell,t} + r_{\text{eval}}^{\ell,t} f_{\text{attn}}^\ell(\mathbf{X}^{\ell,t}) \\ \mathbf{X}^{\ell+1,t} &= \mathbf{Z}^{\ell,t} + r_{\text{eval}}^{\ell,t} f_{\text{ffn}}^\ell(\mathbf{Z}^{\ell,t}) \end{aligned} \qquad (7)$$

## 5. Hyperparameters

**Background**  To avoid the "hyperparameter lottery" phenomenon (Dey et al., 2024), and to ensure we compare against a strong baseline, we systematically optimize learning rate, batch size, and weight decay for each dropout rate before evaluating configurations. Prior literature offers varying strategies—from coupling dropout with $max$-norm regularization (Hinton et al., 2012) to using learning rates $10\times$ larger than baselines (Zhang & He, 2020)—yet systematic consensus remains elusive. To the best of our knowledge, this is the first study to perform joint optimization of these hyperparameters for layer-wise dropout. We tune a small model with dimensions depth $L_{\text{base}}$, width $d_{\text{base}}$ on dataset $D_{\text{base}}$ to determine learning rate $\eta_{\text{base}}$, weight decay $\lambda_{\text{base}}$, initialization $\sigma_{\text{base}}$, and batch size $B_{\text{base}}$, then scale via $\mu$P (Yang & Hu, 2021), CompleteP (Dey et al., 2025), and Power Lines (Bergsma et al., 2025a).

**Dropout Scale**  The scaling parameters $r_{\text{train}}$ and $r_{\text{eval}}$ from Equations 2 and 4 require careful consideration. We define layer density $\rho = 1 - p$. Implementation varies across frameworks: original dropout paper used $r_{\text{train}} = 1$, $r_{\text{eval}} = \rho$ (Hinton et al., 2012); PyTorch/TensorFlow use $r_{\text{train}} = 1/\rho$, $r_{\text{eval}} = 1$; Huang et al. (2016) used $r_{\text{train}} = 1$, $r_{\text{eval}} = \rho$; some libraries set both to 1. We demonstrate that selecting $r_{\text{train}} = 1/\rho$ is critical for stable hyperparameter transfer.

To determine the optimal scale factor $r_{\text{train}}$, we follow CompleteP's Maximal Residual Stream Update Desideratum (Dey et al., 2025), which facilitates hyperparameter transfer across model depths $L$.

*Desideratum* 1 (Maximal Residual Stream Update). Each residual block's weights should contribute order $1/L$ to feature movements, and each non-residual block should contribute constant order. More precisely, for all $\ell \in [L-1]$, each block's parameter update $\boldsymbol{\theta}^\ell \mapsto \boldsymbol{\theta}^\ell + \Delta\boldsymbol{\theta}^\ell$ should contribute the change $\frac{1}{N}\|\Delta_{\boldsymbol{\theta}^\ell}\mathbf{H}^{\ell+1}\|_2^2 \in \Theta(1/L)$. Moreover, for the embedding and unembedding layers we require $\frac{1}{N}\|\Delta\mathbf{W}^0\mathbf{X}\|_2^2 \in \Theta(1)$ and $\frac{1}{N}\|\Delta\mathbf{W}^L\mathbf{H}^L\|_2^2 \in \Theta(1)$.

Since layer dropout reduces the effective depth of the network during training, we treat models with different dropout rates $\rho$ as having different effective depths, and apply this desideratum to ensure stable initialization across these effective depths. Our coordinate checks in Fig. 1 empirically evaluate which scaling factor better satisfies stable initialization across dropout rates: $r_{\text{train}} = 1$ fails, necessitating per-rate tuning, whereas $r_{\text{train}} = 1/\rho$ largely satisfies these

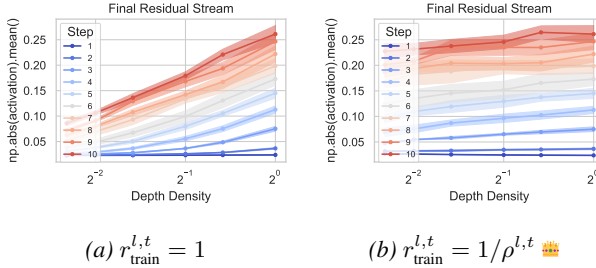

*(a)* $r_{\text{train}}^{l,t} = 1$    *(b)* $r_{\text{train}}^{l,t} = 1/\rho^{l,t}$ 🎉

*Figure 1.* Coordinate Check. Scaling with $r_{\text{train}}^{l,t} = 1/\rho^{l,t}$ during training with layer dropout yields stable activation scale across depth dens...

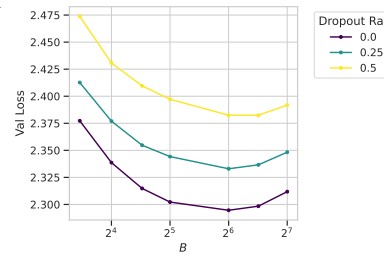

*(a)* Batch size.

*Figure 2.* Analysis of hyperparameter transferability on 271M model. We observe that optimal value for each hyperparameter remains similar across most layer dropout rates with the scaling factor $r_{\text{train}} = 1/\rho$.

checks, enabling optimal hyperparameters transfer across many layer dropout rates.

**Transfer Test**  Fig. 2 verifies that $r_{\text{train}} = 1/\rho$ enables hyperparameter transfer: optimal $B$ remains constant across dropout rates. Hence, we adopt Table A.2's transfer rules with $r_{\text{train}} = 1/\rho$ for all our upcoming experiments. We set $r_{\text{eval}} = 1$ to ensure that $\mathbf{H}_{\text{eval}}^{\ell+1} = \mathbb{E}[\mathbf{H}_{\text{train}}^{\ell+1}]$.

# 6. Dropout Granularity

## 6.1. Model Granularity

**Background**  When layer dropout was first introduced by (Huang et al., 2016), it was applied on residual blocks in CNNs, where each residual block consisted of two convolution-batchnorm pairs, separated by ReLU. In transformers, each layer consists of 2 residual blocks: an attention residual block followed by a FFN residual block. An open question is whether to apply layer dropout separately to attention and FFN (i.e., the Bernoulli mask tensors $\mathbf{M}_{\text{attn}}^{\ell,t}$ and $\mathbf{M}_{\text{ffn}}^{\ell,t}$ are sampled independently at each training step, $t$), which we refer to as Sub-Layer Dropout, or to apply it on the whole transformer layer (i.e., $\mathbf{M}_{\text{attn}}^{\ell,t} = \mathbf{M}_{\text{ffn}}^{\ell,t} \quad \forall \ell$), which we refer to as Layer Dropout. Different research work have used different types: DINOv2 (Oquab et al., 2024) and (Zhang & He, 2020), used sub-layer dropout, while LayerDrop (Fan et al., 2020) and LayerSkip (Elhoushi

et al., 2024) used layer dropout. However, to the best of our knowledge, we are the first to systematically evaluate a comparison between them.

**Analysis**  In Table 1 we compare layer dropout and sub-layer dropout at various model sizes. The results clearly show that in terms of accuracy, Layer Dropout is better. Note that as model size increases, loss degradation introduced by dropout diminishes, which will later encourage us to try larger dropout rates for larger models.

*Table 1.* Ablating model granularities. Models trained at 20 TPP.

| Model Size | Dropout Type | Dropout Rate | Train Loss ↓ | % Δ ↓ | Val Loss ↓ | % Δ ↓ |
|---|---|---|---|---|---|---|
| 271M | Baseline | — | 2.293 | 0.00% | 2.294 | 0.00% |
| | SubLayer | 0.1 | 2.421 | 4.72% | 2.377 | 3.61% |
| | Layer 🎉 | 0.1 | **2.419** | 3.95% | **2.367** | 3.18% |
| 503M | Baseline | — | 2.140 | 0.00% | 2.110 | 0.00% |
| | SubLayer | 0.1 | 2.260 | 3.94% | 2.174 | 3.03% |
| | Layer 🎉 | 0.1 | **2.178** | 2.60% | **2.170** | 2.84% |

> **Finding 1**: Layer dropout that drops whole transformer blocks for each sample, leads to higher accuracy results than sub-layer dropout that drops attention and FFN sub-blocks separately.

## 6.2. Tensor Granularity

**Background**  The next open question we tackle is whether it is better to apply layer dropout at batch granularity (i.e., $\mathbf{M}^{\ell,t}[b,:,:]$ takes the same value across all sequences $b$) or at sequence granularity (i.e., $\mathbf{M}^{\ell,t}[b,:,:]$ is sampled independently for each sequence $b$). In literature, this does not seem to have been discussed, and we usually need to resort to the codebases of different papers to find out which type each has used. The implementation of the pioneer Stochastic Depth paper[2] as well as the `fairseq`[3] implementation of Layer-Drop used per-batch layer dropout, while DINOv2 (Oquab et al., 2024)[4], `timm`[5], and `torchtune`[6] implementation of LayerSkip used per-sequence. To the best of our knowledge, we are the first to systematically compare per-batch and per-sequence layer dropout.

---

[2]`https://github.com/yueatsprograms/Stochastic_Depth/blob/master/ResidualDrop.lua`

[3]`https://github.com/facebookresearch/fairseq/blob/main/fairseq/modules/layer_drop.py`

[4]`https://github.com/facebookresearch/dinov2/blob/main/dinov2/layers/drop_path.py`

[5]`https://github.com/huggingface/pytorch-image-models/blob/main/timm/layers/drop.py`

[6]`https://github.com/meta-pytorch/torchtune/blob/main/torchtune/modules/layer_dropout.py`

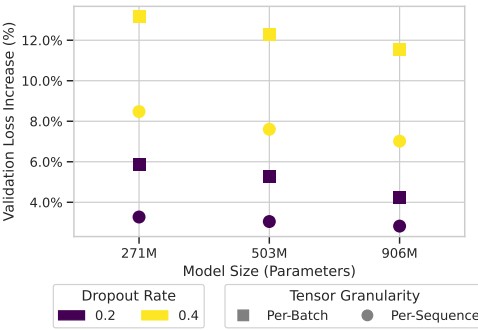

*Figure 3.* Ablating tensor granularity. Models trained at 20 TPP.

**Analysis** Fig. 3 compares the accuracy results of applying layer dropout per batch and per sequence. The results clearly show that per-sequence leads to better losses. This is in line with the notion that finer grain sparsity leads to higher accuracy.

> **Finding 2**: For any given layer dropout rate, dropout per-sequence leads to lower loss than dropout per-batch.

## 7. Dropout Configurations

### 7.1. Dropout Distribution

**Background** Various dropout distributions across layers have been proposed to optimize training efficiency and model depth. We formalize three primary distributions for dropout rate $p$ at layer $\ell$:

1. **Uniform Distribution:** $p_{\text{uniform}}^{\ell,t} = p_{\max}$.
2. **Increasing Layer Distribution (ILD):** $p_{\text{ILD}}^{\ell,t} = \frac{\ell}{L-1} \cdot p_{\max}$ (Huang et al., 2016; Oquab et al., 2024; Zhang & He, 2020; Elhoushi et al., 2024).
3. **Alternating Layer Distribution (ALD):** $p_{\text{ALD}}^{\ell,t} = p_{\max} \cdot \mathbf{1}_{\ell \equiv 1 (\text{mod } 2)}$ (Fan et al., 2020).

where $p_{\max}$ is the maximum dropout rate. Note that $\text{unique}(p_{\text{ALD}}^{\ell,t}) \in \{0, p_{\max}\}$, whereas $p_{\text{ILD}}^{\ell,t} \in [0, p_{\max}]$.

For any distribution, the average dropout and corresponding nonembedding FLOPs[7] savings at step $t$ are defined as:

$$p_{\text{mean}}^t = \text{FLOPs Savings}^t = \frac{1}{L} \sum_{\ell=0}^{L-1} p^{\ell,t} \qquad (8)$$

Mathematically, $p_{\text{ILD}_{\text{mean}}}^t = 0.5 p_{\max}$ and $p_{\text{ALD}_{\text{mean}}}^t = \frac{\lfloor L/2 \rfloor}{L} p_{\max}$, which is $\approx 0.5 p_{\max}$ for typical $L$. To the best of our knowledge, this study is the first to systematically analyze the differences between these layer dropout distributions at fixed FLOPs budgets.

---

[7]For the remaining of the paper, we use the term FLOPs to refer to nonembedding FLOPs.

**Analysis** In Table 2, we compare uniform, ILD, and ALD grouped by equivalent FLOPs savings, finding that non-uniform distributions consistently outperform uniform ones under a fixed budget. While ALD is superior at the smallest model size, its advantage diminishes with scale, whereas ILD's improvement over uniform widens. Although our ALD results with $p_{\max} = 0.2$ do not beat the baseline as reported in the multi-epoch regime of LayerDrop (Fan et al., 2020), the observed reduction in dropout-induced degradation as models grow encourages further investigation at larger scales.

*Table 2.* Analysis of Dropout Distributions across layers. Models trained at 20 TPP.

| Model | Training FLOPs Savings | Max Rate | Dropout Distb. | Val | % Δ |
|---|---|---|---|---|---|
| 271M | 0% | - | - | 2.294 | 0.00% |
| | 10% | 0.1 | Uniform | 2.331 | 1.62% |
| | | 0.2 | Alternating | **2.323** | **1.29%** |
| | | 0.2 | Increasing | 2.328 | 1.52% |
| | 20% | 0.2 | Uniform | 2.369 | 3.27% |
| | | 0.4 | Alternating | **2.357** | **2.76%** |
| | | 0.4 | Increasing | 2.363 | 3.01% |
| 503M | 0% | - | - | 2.110 | 0.00% |
| | 10% | 0.1 | Uniform | 2.141 | 1.48% |
| | | 0.2 | Alternating | 2.143 | 1.57% |
| | | 0.2 | Increasing | **2.132** | **1.08%** |
| | 20% | 0.2 | Uniform | 2.174 | 3.05% |
| | | 0.4 | Alternating | 2.169 | 2.82% |
| | | 0.4 | Increasing | **2.162** | **2.48%** |
| 906M | 0% | - | - | 1.953 | 0.00% |
| | 10% | 0.1 | Uniform | 1.977 | 1.26% |
| | | 0.2 | Alternating | 1.979 | 1.36% |
| | | 0.2 | Increasing | **1.972** | **1.01%** |
| | 20% | 0.2 | Uniform | 2.008 | 2.82% |
| | | 0.4 | Alternating | 2.004 | 2.66% |
| | | 0.4 | Increasing | **1.998** | **2.34%** |

> **Finding 3**: For the same training FLOPs budget, non-uniform dropout distribution across layers is better than uniform. As a model scales, ILD is recommended.

### 7.2. Dropout Schedule

**Background** While distributions govern sparsity across depth, the temporal schedule determines how regularization pressure evolves throughout pre-training. Hillier et al. (2024) found decreasing schedules were better for LLM pre-training, whereas increasing schedules were superior for fine-tuning; however, Liu et al. (2025) recently claimed both fail in modern regimes. We formalize various time schedules for dropout rate $p$ at step $t$ over total duration $T$, where $p_{\text{dist}}^\ell$ represents a chosen layer distribution:

1. **Constant Time Schedule:** $p_{\text{constant}}^{\ell,t} = p_{\text{dist}}^\ell$
2. **Increasing Time Schedule (ITS):** $p_{\text{ITS}}^{\ell,t} = p_{\text{dist}}^\ell \cdot \left( \frac{t}{T-1} \right)$
3. **Decreasing Time Schedule (DTS):** $p_{\text{DTS}}^{\ell,t} = p_{\text{dist}}^\ell \cdot$

$$\left(1 - \frac{t}{T-1}\right)$$

To compare these schedules fairly, we define the mean training dropout $\bar{P}$ as the average rate across depth and time, representing total active training FLOPs savings:

$$\bar{P} = \text{FLOPs Savings}_{\text{total}} = \frac{1}{T}\sum_{t=0}^{T-1}\left(\frac{1}{L}\sum_{\ell=0}^{L-1} p^{\ell,t}\right) \quad (9)$$

Consequentially, $\bar{P}_{\text{uniform,ITS}} = 0.5 p_{\max}$ and $\bar{P}_{\text{ILD,ITS}} = \bar{P}_{\text{ILD,DTS}} = 0.25 p_{\max}$.

**Analysis**   In Table 3, we group configurations by total FLOPs savings. Across all scales, decreasing schedules consistently outperforms constant and increasing schedules. Notably, at 5% FLOPs savings for 503M & 906M, combined ILD and DTS achieves lower or almost equivalent validation losses than the dense baseline, demonstrating for the first time that it is possible to beat the dense baseline with fewer training FLOPs.

Conversely, increasing schedule significantly degrades loss. While Zhang & He (2020) reported positive results with an exponential increasing schedule, we do not observe benefits in our large-scale single-epoch regime. We hypothesize the decreasing schedule's effectiveness stems from high initial noise forcing weight space exploration (reducing bias), while subsequent decay allows settling into a stable minimum (reducing variance). This can also be viewed as a form of stochastic model growing, where effective capacity increases smoothly without explicit re-initialization of conventional model growing (e.g., (Samragh et al., 2024)). It can also be viewed as a form of curriculum learning (Wang et al., 2021), that starts training with a hard task of learning using small effective depth and gradually becomes easier as effective depth increases.

We leave schedules such as applying dropout to mid-training, SFT, or continual pre-training for future work.

> **Finding 4**: For a fixed training FLOPs budget, a schedule decaying from a maximum rate to zero consistently achieves the highest accuracy, outperforming both constant and increasing schedules across all scales.

> **Key takeaway 1**: The best recommended practice for layer dropout configuration is increasing dropout across layers and decreasing dropout across time.

## 8. Inference Optimizations

A primary motivation for pre-training with layer dropout is to induce robustness to depth-wise optimizations, including early exit, layer skipping, and layer pruning. We explore techniques that keep pre-trained weights intact. We leave pruning approaches that require fine-tuning or modifying

weights, e.g., (Xia et al., 2024; Lu et al., 2024), for future work.

### 8.1. Zero-Shot Inference Benefits

Here we cover merely applying autoregressive decoding on a subset of the model's layers without any modification.

#### 8.1.1. EARLY EXIT

We define early exit at layer $\ell'$ as executing the embedding layer, transformer layers 0 to $\ell'-1$, and the unembedding layer. This "static early exit" is equivalent to skipping layers $\ell'$ to $L-1$. Fig. 4 illustrate results for various models. We observe that models trained without dropout exhibit significant loss deterioration when exiting early, whereas dropout-trained models maintain steady loss across a portion of layers, with higher dropout rates yielding lower losses.

Does a decreasing schedule sacrifice depth robustness by ending training dropout-free? The results say no: training with a decreasing schedule, e.g., 0.4 (ILD,DTS), significantly outperforms a zero-dropout baseline at early exit and matches a constant, e.g., 0.2 (ILD), when controlling for training FLOPs — proving that exposure to dropout during early training leaves lasting benefits.

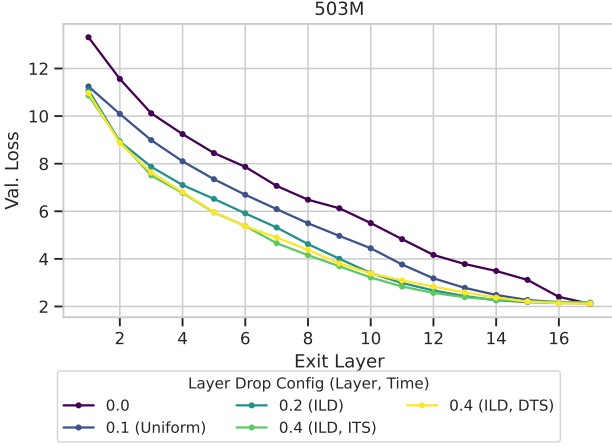

*Figure 4.* Early-exit validation loss for 270M model at 20 TPP: no dropout vs. dropout configurations with 20% FLOPs savings.

#### 8.1.2. INTERMEDIATE LAYER SKIPPING

Layer dropout induces structural robustness enabling models to function when layers are skipped at inference. As shown in Fig. A.2a, dense baselines exhibit immediate loss spikes when any layers are skipped, whereas ALD facilitates graceful degradation. This zero-shot "elastic" effect allows a 906M model to bridge the gap toward smaller dense baselines, as shown in Fig. A.2b, providing flexibility — typically requiring complex modifications and/or continual pretraining like LlamaFlex (Cai et al., 2025) or Flextron (Cai

| Training FLOPs Savings | Max Dropout | Layer Dist. | Time Sched. | 271M Val | 271M % Δ | 503M Val | 503M % Δ | 906M Val | 906M % Δ |
|---|---|---|---|---|---|---|---|---|---|
| **0%** | – | – | – | 2.294 | 0.00% | 2.110 | 0.00% | 1.953 | 0.00% |
| **5%** | 0.05 | – | – | 2.311 | 0.77% | 2.124 | 0.67% | 1.963 | 0.54% |
| | 0.1 | Increasing | – | 2.310 | 0.73% | 2.121 | 0.56% | 1.961 | 0.42% |
| | 0.2 | Increasing | Increasing | 2.325 | 1.36% | 2.139 | 1.38% | 1.979 | 1.34% |
| | 0.2 | Increasing | Decreasing 🏆 | **2.304** | **0.46%** | **2.110** | **0.03%** | **1.951** | **−0.06%** |
| **10%** | 0.1 | – | – | 2.331 | 1.62% | 2.141 | 1.48% | 1.977 | 1.26% |
| | 0.2 | Increasing | – | 2.328 | 1.52% | 2.132 | 1.08% | 1.972 | 1.01% |
| | 0.4 | Increasing | Increasing | 2.357 | 2.75% | 2.172 | 2.95% | 2.014 | 3.15% |
| | 0.4 | Increasing | Decreasing 🏆 | **2.312** | **0.82%** | **2.116** | **0.28%** | **1.955** | **0.10%** |
| **20%** | 0.2 | – | – | 2.369 | 3.27% | 2.174 | 3.05% | 2.008 | 2.82% |
| | 0.4 | Increasing | – | 2.363 | 3.01% | 2.162 | 2.48% | 1.998 | 2.34% |
| | 0.8 | Increasing | Increasing | 2.442 | 6.46% | 2.257 | 6.99% | 2.090 | 7.04% |
| | 0.8 | Increasing | Decreasing 🏆 | **2.361** | **2.93%** | **2.147** | **1.75%** | **1.983** | **1.55%** |

*Table 3.* Ablating dropout time schedule. We group different dropout configurations that have the same active non-embedding FLOPs reduction induced by layer dropout across all training steps. Models trained at 20 TPP.

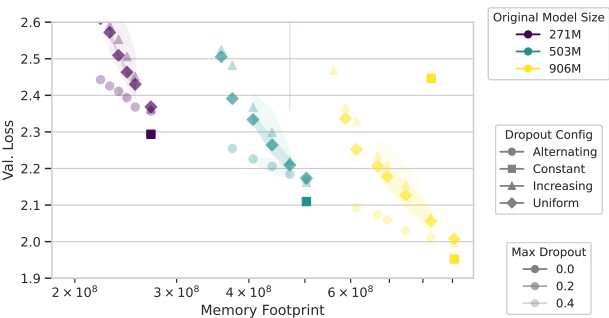

*Figure 5.* Intermediate layer skipping loss for models at 20 TPP: baseline vs. dropout configurations with 20% FLOPs savings. Extended results in Fig. A.2.

et al., 2024) — "for free" within the standard pre-training recipe.

Ablations at $p_{max} = 0.2$ show that while all dropout variants improve skip-robustness, ALD offers superior retention for non-contiguous skipping (Fig. A.2c). Under iso-FLOP conditions (Fig. 5), ALD maintains lower validation loss than ILD for an equivalent 20% training compute reduction, confirming ALD as optimal for depth-wise inference elasticity at a fixed budget.

A clear trade-off emerges: ALD excels at skip-robustness but fails at early-exit, while ILD achieves better base accuracy and early-exit robustness with sub-optimal skip capability. Practitioners should choose based on deployment needs, and consider ALD with extended training to recover baseline accuracy.

> **Finding 5**: Average training dropout rate predicts zero-shot robustness to early exit and layer skipping without retraining.

## 8.2. Post-Training Inference Benefits

While zero-shot techniques exploit the inherent redundancy of a model, further efficiency gains can be achieved through targeted post-training modifications that do not alter the pre-trained weights. We define post-training benefits as those derived from secondary training phases—such as continual pre-training or fine-tuning—specifically focused on optimizing inference throughput. In this work, we limit our investigation to "weight-frozen" methods where the transformer backbone remains static, and optimization is achieved by training auxiliary modules like adapters or routers. This paradigm ensures that the model's foundational knowledge is preserved while expanding the Pareto-optimal frontier of its depth-wise flexibility. Here we cover using adapters and self-speculative decoding, and leave using routers (such as in (Jiang et al., 2024)) for future work.

### 8.2.1. EARLY EXIT ADAPTERS

**Background** To evaluate if the structural benefits of layer dropout persist after supervised optimization, we utilize the *Balcony* framework for depth-based dynamic inference (Jamialahmadi et al., 2025). Balcony is a lightweight approach that freezes the pre-trained backbone and inserts additional transformer layers as "exit adapters" at selected points. These adapters are trained using a self-distillation objective where a Kullback–Leibler (KL) divergence loss aligns intermediate sub-model outputs with the final layer's predictions. While Jamialahmadi et al. (2025) demonstrates that incorporating these adapters directly into the pre-training phase leads to even lower early-exit loss compared to post-training addition, such joint training increases the memory footprint and FLOPs per step, potentially slowing down the pre-training process.

**Analysis** As shown in Fig. 6, models pre-trained with layer dropout consistently outperform dense baselines

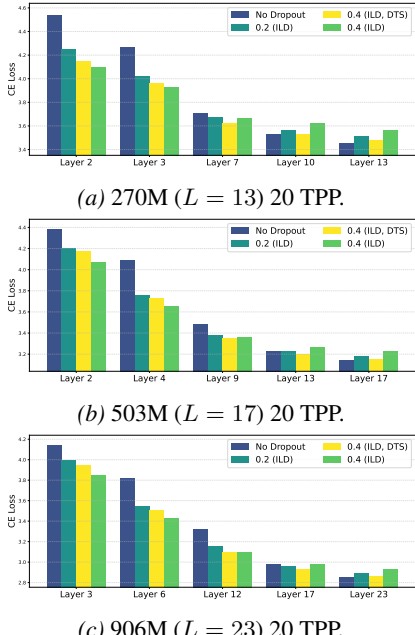

*(a)* 270M ($L = 13$) 20 TPP.

*(b)* 503M ($L = 17$) 20 TPP.

*(c)* 906M ($L = 23$) 20 TPP.

*Figure 6.* Early exit losses for models pre-trained with different dropout configurations, followed by freezing their weights and training early exit adapters as proposed by Balcony (Jamialahmadi et al., 2025). Models pre-trained with dropout always lead to better early exit losses even after adding exit adapters.

across all model scales—270M, 503M, and 906M—even when exit adapters are only added post-training. Our proposed method offers a more efficient alternative to joint adapter pre-training: by incorporating layer dropout, we improve the loss of earlier layers without increasing the memory footprint or computational overhead. In fact, layer dropout actively reduces training FLOPs while inducing a permanent structural robustness that auxiliary training can leverage but cannot fully replicate on a standard dense model. Furthermore, models trained with higher dropout rates demonstrate a superior "head-start" for adapter training, reaching lower validation losses at earlier layers than their low-dropout counterparts, confirming that depth-aware pre-training is a prerequisite for maximizing the efficacy of post-training strategies.

### 8.2.2. SELF-SPECULATIVE DECODING

**Background** Speculative decoding accelerates autoregressive inference by using a fast "draft" model to predict tokens that are validated in parallel by a larger "target" model, enabling lossless speedup (Leviathan et al., 2023). Self-speculative methods, such as *Draft & Verify*, use a subset of the target model's own layers to act as the drafter (Zhang et al., 2024a). The effectiveness of this approach depends on identifying a layer subset that is small enough for high throughput yet accurate enough to maintain high token acceptance rates; otherwise, drafting overhead may exceed the

verification savings.

**Analysis** We hypothesized that the structural elasticity induced by layer dropout enables the discovery of more efficient subsets. While Zhang et al. (2024a) used Bayesian optimization to find draft layers, we also apply other search methods: genetic algorithms, hill climbing, and simulated annealing, and select the search result that leads to highest speedup. Results in Table 4 confirm this showing models pre-trained with higher layer dropout obtain higher speedups during self-speculative decoding. For such models, search methods are able to find a subset of layers that achieve better trade offs of acceptance rate, $\alpha$, and draft decoding time, $T_{\text{Draft}}$. We leave for future work the evaluation of other self-speculative techniques like LayerSkip (Elhoushi et al., 2024), which uses early exit for drafting, and Kangaroo (Liu et al., 2024), which employs early exit with adapters trained via Balcony (Jamialahmadi et al., 2025) as described in Sec. 8.2.1.

| Model Size | TPP | Dropout Config. | Self-Speculative Decoding | | |
|---|---|---|---|---|---|
| | | | Speedup | $\alpha$ | $\frac{T_{\text{Draft}}}{T_{\text{Target}}}$ |
| 270M | 20 | 0 | $1.01\times$ | 92% | 0.65 |
| | | 0.2 (ILD) | $1.20\times$ | 86% | 0.55 |
| | | 0.4 (ILD) | $\mathbf{1.35\times}$ | 89% | 0.48 |
| | | 0.4 (ILD, DTS) | $1.22\times$ | 73% | 0.57 |
| 504M | 20 | 0 | $1.03\times$ | 90% | 0.66 |
| | | 0.2 (ILD) | $1.22\times$ | 89% | 0.53 |
| | | 0.4 (ILD) | $\mathbf{1.43\times}$ | 97% | 0.42 |
| | | 0.4 (ILD, DTS) | $1.20\times$ | 90% | 0.52 |
| 906M | 20 | 0 | $1.06\times$ | 94% | 0.66 |
| | | 0.2 (ILD) | $1.32\times$ | 98% | 0.49 |
| | | 0.4 (ILD) | $\mathbf{1.40\times}$ | 97% | 0.48 |
| | | 0.4 (ILD, DTS) | $1.29\times$ | 95% | 0.52 |

*Table 4.* Self-speculative decoding speedup across model scales using Draft & Verify (Zhang et al., 2024a). $\alpha \in [0, 1]$ is acceptance rate for draft length $\gamma = 5$, $T_{\text{Draft}}$ is time to decode a single token for the selected subset of layers, and $T_{\text{Target}}$ for the full model.

## 9. Scaling Analysis

To evaluate whether layer dropout benefits persist beyond compute-optimal regimes, we extend our analysis to high tokens-per-parameter (TPP) budgets, mirroring established scaling law frameworks.

Our recommended ILD+DTS configuration demonstrates that absolute degradation remains remarkably stable as shown in Fig. 7, typically staying within $\approx 0.50\%$ of the baseline even at high TPP. These results indicate that layer dropout does not impede scaling performance. Instead, the stability observed suggests a robust, compute-efficient pre-training pathway that maintains structural benefits as data scales to training budgets of typical modern foundational LLMs.

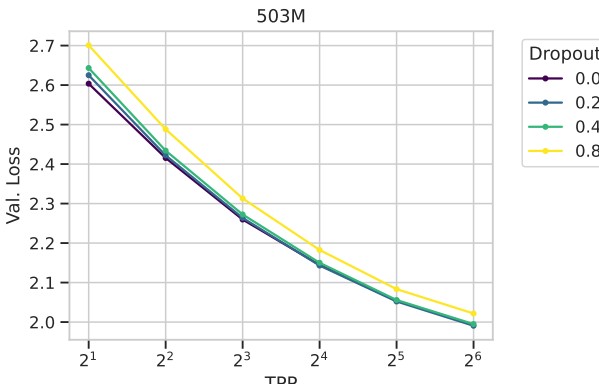

*Figure 7.* Validation loss across TPP for ILD with DTS, showing competitive performance with dense baselines as tokens-per-parameter increase.

## 10. Conclusion

While researchers have abandoned even small dropout rates for pretraining LLMs, our study has presented that LLMs can achieve resilience to aggressive dropout rates, maintaining baseline accuracy while yielding up to 20% training FLOPs savings. This depth-aware training induces structural elasticity that bridges performance gaps between discrete model sizes and unlocks inference speedups of up to $1.4\times$ through strategies like self-speculative decoding. These findings encourage the re-adoption of layer dropout as a non-invasive, foundational component for efficient and elastic large-scale training. Despite these gains, our work has limitations—including the lack of comparisons to learned depth mechanisms (Raposo et al., 2024)—which we detail in Sec. D.

## Impact Statement

This paper presents work whose goal is to advance the field of machine learning. There are many potential societal consequences of our work, none of which we feel must be specifically highlighted here.

## Acknowledgments

We would like to thank Shaheer Mohammad and Sam McPhail for infrastructure support at Cerebras.

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

# Appendix

## Glossary

**Alternating Layer Distribution (ALD)** A dropout distribution where dropout is not applied on the first layer, but is applied on alternating layers proceeding that. It is expressed mathematically as: $p_{\text{ALD}}^{\ell,t} = p_{\max} \cdot \mathbf{1}_{\ell \equiv 1 (\text{mod } 2)}$.

**Constant Time Schedule** A temporal schedule for layer dropout where the dropout rate remains constant throughout training. It is expressed mathematically as $p_{\text{const}}^{\ell,t} = p_{\text{dist}}^{\ell}$.

**Decreasing Time Schedule (DTS)** A temporal schedule for layer dropout where the dropout rate is at its maximum at the start of pre-training and decays linearly to zero by the final training step. It is expressed mathematically as $p_{\text{DTS}}^{\ell,t} = p_{\text{dist}}^{\ell} \cdot \left(1 - \frac{t}{T}\right)$. In this work, we demonstrate that this schedule helps stabilize early training while allowing the model to settle into a dense state for final convergence.

**FFN** Feed Forward Networks.

**Increasing Layer Distribution (ILD)** A layer dropout configuration where the dropout rate starts at 0 for the first layer, i.e., $p^{\ell=0} = 0$, and linearly increases across layers to a maximum dropout rate, i.e., $p^{\ell=L-1} = p_{\max}$. It is mathematically expressed as: $p_{\text{ILD}}^{\ell,t} = \frac{\ell}{L-1} \cdot p_{\max}$.

**Increasing Time Schedule (ITS)** A temporal schedule for layer dropout where the dropout rate starts at zero and increases linearly to its maximum by the final training step. It is expressed mathematically as $p_{\text{ITS}}^{\ell,t} = p_{\text{dist}}^{\ell} \cdot \left(\frac{t}{T}\right)$.

**Layer Dropout** A form of dropout where entire layers of a neural network (e.g., transformer blocks) are randomly skipped during training. Unlike standard dropout, which zeroes out individual activations, layer dropout operates at the structural level, reducing the effective depth of the network on each forward pass. In the context of transformers, we use this term to indicate applying layer dropout on the granularity of a whole transformer block.

**Stochastic Depth** An alternative term for Layer Dropout.

**Sub-Layer Dropout** In the context of transformers, refers to applying layer dropout on attention and FFN blocks independently.

**Uniform Distribution** A layer dropout configuration where dropout rates of all layers are set to the same value, and can be mathematically expressed as $p_{\text{uniform}}^{\ell,t} = p_{\max}$.

# A. Experimental Settings

| Model | Hidden Dim. $D$ | Layers $L$ | Heads | Head Size | FFN Mult. |
|-------|-----------------|------------|-------|-----------|-----------|
| 271M | 640 | 13 | 10 | 64 | 8 |
| 504M | 896 | 17 | 14 | 64 | 8 |
| 906M | 1152 | 23 | 9 | 128 | 8 |
| 1.8B | 1536 | 30 | 12 | 128 | 8 |
| 3.9B | 2048 | 40 | 16 | 128 | 8 |
| 8.2B | 2688 | 52 | 21 | 128 | 8 |

*Table A.1.* Model architectures used in the experiments.

*Table A.2.* Summary of SP, μP, and CompleteP with layer dropout rate for a transformer model. Terms related to width(introduced by μP (Yang & Hu, 2021)), depth(introduced by CompleteP (Dey et al., 2025)), data size(introduced by (Bergsma et al., 2025a)), and dropout (introduced in this paper) controls are highlighted in orange, green, brown, and purple respectively. Additional tunable parameters are highlighted in blue. *Hidden* refers to all linear layers in the transformer backbone. Layer density, $\rho$ is the complement of layer dropout, $p$ such that $\rho = 1 - p$ .

| Parameterization | Base Training | Scaled Training |
|------------------|---------------|-----------------|
| Width | $d_{\text{base}}$ | $d_{\text{base}} \cdot m_d$ |
| Depth | $L_{\text{base}}$ | $L_{\text{base}} \cdot m_L$ |
| Dataset Size | $D_{\text{base}}$ | $D_{\text{base}} \cdot m_D$ |
| Layer Density | 1 | $\rho$ |
| Model Size | $N_{\text{base}} = g(d_{\text{base}}, L_{\text{base}})$ | $N = g(d_{\text{base}} \cdot m_d, L_{\text{base}} \cdot m_L)$ |
| Tokens per Parameter | $\text{TPP}_{\text{base}} = D_{\text{base}}/N_{\text{base}}$ | $\text{TPP} = D_{\text{base}} \cdot m_D/N$ |
| Batch Size | $B_{\text{base}}$ | $B_{\text{base}} \cdot m_D^{0.4}$ |
| Timescale (AdamW) | $\tau_{\text{EMA}_{\text{base}}}$ | $\tau_{\text{EMA}} = \tau_{\text{EMA}_{\text{base}}} \cdot \left(\frac{\text{TPP}}{\text{TPP}_{\text{base}}}\right)^{-0.5}$ |
| Learning Rate Warmup | min(10% of total tokens, 375M tokens) | min(10% of total tokens, 375M tokens) |
| Learning Schedule | Linear Decay-to-Zero | Linear Decay-to-Zero |
| Emb. Init. Var. | $\sigma_{\text{base}}^2$ | $\sigma_{\text{base}}^2$ |
| Emb. LR (AdamW) | $\eta_{\text{base}}$ | $\eta_{\text{base}}$ |
| Pre-LN Init. Var. | $\sigma_{\text{base}}^2$ | $\sigma_{\text{base}}^2$ |
| Pre-LN LR (AdamW) | $\eta_{\text{base}}$ | $\eta_{\text{base}}$ |
| Hidden Init. Var. | $\sigma_{\text{base}}^2$ | $\sigma_{\text{base}}^2 \cdot m_d^{-1}$ |
| Hidden LR (AdamW) | $\eta_{\text{base}}$ | $\eta_{\text{base}} \cdot m_d^{-1}$ |
| Hidden Bias LR (AdamW) | $\eta_{\text{base}}$ | $\eta_{\text{base}}$ |
| Hidden WD (AdamW) | $\frac{B_{\text{base}}}{\eta_{\text{base}} \tau_{\text{EMA}_{\text{base}}} D_{\text{base}}}$ | $\frac{B_{\text{base}}}{\eta_{\text{base}} \tau_{\text{EMA}} D_{\text{base}} m_D^{0.4}} \cdot m_d$ |
| Attention Residual | $\mathbf{X}^l + f_{\text{attn}}(\mathbf{X}^l)$ | $\mathbf{X}^l + m_L^{-1}\rho^{-1} \cdot f_{\text{attn}}(\mathbf{X}^l)$ |
| FFN Residual | $\mathbf{Z}^l + f_{\text{ffn}}(\mathbf{Z}^l)$ | $\mathbf{Z}^l + m_L^{-1}\rho^{-1} \cdot f_{\text{ffn}}(\mathbf{Z}^l)$ |
| Final-LN Init. Var. | $\sigma_{\text{base}}^2$ | $\sigma_{\text{base}}^2$ |
| Final-LN LR (AdamW) | $\eta_{\text{base}}$ | $\eta_{\text{base}}$ |
| Unemb. Init. Var. | $\sigma_{\text{base}}^2$ | $\sigma_{\text{base}}^2$ |
| Unemb. LR (AdamW) | $\eta_{\text{base}}$ | $\eta_{\text{base}}$ |
| Unemb. Fwd. | $\mathbf{X}^L \mathbf{W}_{\text{unemb}}^{\top}$ | $\mathbf{X}^L \mathbf{W}_{\text{unemb}}^{\top} \cdot m_d^{-1}$ |
| AdamW $\epsilon$ (Residual blocks) | $\epsilon_{\text{base}}$ | $\epsilon_{\text{base}} \cdot m_d^{-1} \cdot m_L^{-1}$ |
| AdamW $\epsilon$ (Emb. & Unemb.) | $\epsilon_{\text{base}}$ | $\epsilon_{\text{base}} \cdot m_d^{-1}$ |

# B. Coordinate Check

Fig. 1 and Fig. A.1 show coordinate check plots for uniform and non-uniform dropouts respectively. They show Frobenius norm of activations after merged residual streams from attention and FFN blocks across a 40 layer model after training step $t$, using CompleteP (that scales residuals by $\frac{L_{\text{base}}}{L}$), $r_{\text{train}}^{\ell}$ for layer $\ell$.

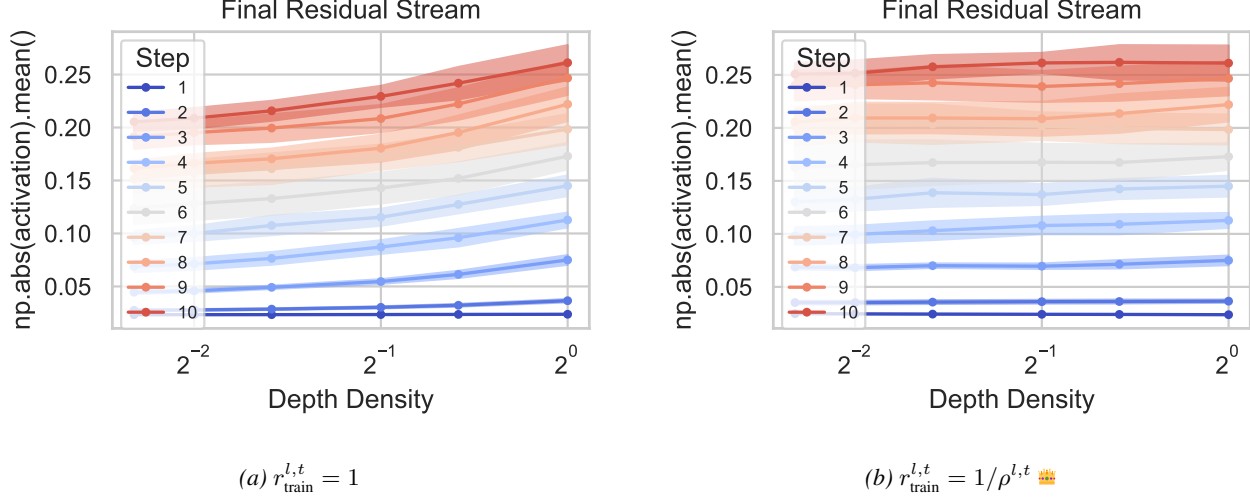

*(a)* $r_{\text{train}}^{l,t} = 1$  *(b)* $r_{\text{train}}^{l,t} = 1/\rho^{l,t}$ 🏆

*Figure A.1.* Coordinate Check passing for non-uniform dropout distribution (ILD).

# C. Additional Results

## C.1. Downstream Tasks

*Table A.3.* Downstream task performance benchmarks.

| | | Training Config. | | Downstream Tasks ↑ | | | | | | | | | | |
|---|---|---|---|---|---|---|---|---|---|---|---|---|---|---|
| **Size** | **TPP** | **Layer Dropout** | **FLOPs Sav. ↑** | **BoolQ** | **PIQA** | **SIQA** | **Hella.** | **Wino.** | **ARC-c** | **ARC-e** | **OBQA** | **Lamb.** | **COPA** | **RACE** |
| 271M | 20 | 0 | 0% | 0.522 | 0.593 | 0.358 | 0.293 | 0.500 | 0.224 | 0.415 | 0.274 | 0.226 | 0.580 | 0.265 |
| 271M | 20 | 0.2 (ILD) | 10% | 0.613 | 0.579 | 0.349 | 0.285 | 0.505 | 0.224 | 0.399 | 0.262 | 0.218 | 0.580 | 0.273 |
| 271M | 20 | 0.4 (ILD, DTS) | 10% | 0.452 | 0.584 | 0.358 | 0.290 | 0.523 | 0.241 | 0.409 | 0.274 | 0.229 | 0.600 | 0.269 |
| 271M | 20 | 0.4 (ILD) | 20% | 0.575 | 0.584 | 0.345 | 0.283 | 0.499 | 0.234 | 0.400 | 0.266 | 0.223 | 0.610 | 0.253 |
| 503M | 20 | 0 | 0% | 0.556 | 0.594 | 0.360 | 0.316 | 0.515 | 0.246 | 0.441 | 0.282 | 0.280 | 0.560 | 0.279 |
| 503M | 20 | 0.2 (ILD) | 10% | 0.550 | 0.592 | 0.357 | 0.306 | 0.497 | 0.234 | 0.446 | 0.272 | 0.290 | 0.620 | 0.266 |
| 503M | 20 | 0.4 (ILD, DTS) | 10% | 0.575 | 0.586 | 0.361 | 0.309 | 0.516 | 0.239 | 0.443 | 0.274 | 0.281 | 0.670 | 0.268 |
| 503M | 20 | 0.4 (ILD) | 20% | 0.573 | 0.596 | 0.358 | 0.301 | 0.499 | 0.241 | 0.434 | 0.270 | 0.283 | 0.640 | 0.263 |
| 906M | 20 | 0 | 0% | 0.475 | 0.611 | 0.360 | 0.353 | 0.493 | 0.242 | 0.493 | 0.286 | 0.337 | 0.660 | 0.286 |
| 906M | 20 | 0.2 (ILD) | 10% | 0.495 | 0.607 | 0.386 | 0.343 | 0.510 | 0.242 | 0.477 | 0.288 | 0.339 | 0.670 | 0.300 |
| 906M | 20 | 0.4 (ILD, DTS) | 10% | 0.566 | 0.609 | 0.365 | 0.353 | 0.527 | 0.257 | 0.484 | 0.272 | 0.345 | 0.680 | 0.305 |
| 906M | 20 | 0.4 (ILD) | 20% | 0.503 | 0.613 | 0.357 | 0.334 | 0.513 | 0.245 | 0.471 | 0.274 | 0.330 | 0.650 | 0.292 |

## C.2. Zero-Shot Inference Benefits

### C.2.1. INTERMEDIATE LAYER SKIPPING

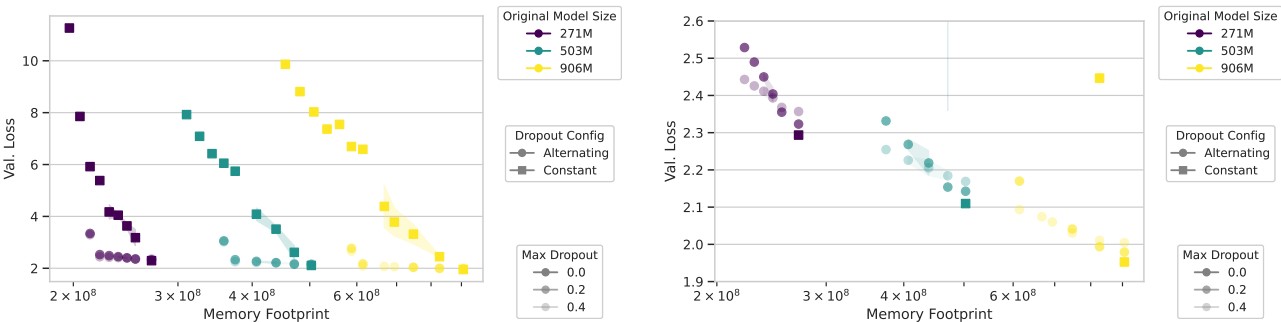

*(a)* Ablating different maximum dropout values for Alternate Distribution.

*(b)* Ablating different maximum dropout values for Alternate Distribution, zooming on skipping configurations with lower losses.

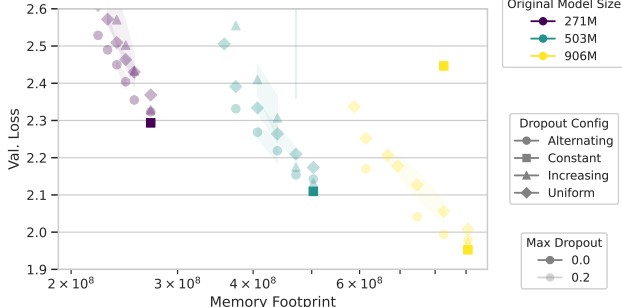

*(c)* Ablating different dropout configurations for the same maximum dropout of 0.2.

*Figure A.2.* Comparison of intermediate layer skipping validation losses for models trained with different dropout configurations. All models trained with 20 TPP.

# D. Limitations

While our results establish a robust framework for layer dropout at scale, our work has multiple limitations:

- **Alternative Granularities:** We focused exclusively on transformer-level dropout to induce depth-wise robustness. Whether other granularities, such as attention-head or neuron-level dropout, can induce similar structural resilience remains an open question.

- **Comparison with Learned Depth Optimization:** This study did not compare structured layer dropout against learned depth-aware mechanisms such as Mixture-of-Depths (Raposo et al., 2024). Investigating the trade-offs between stochastic layer removal and dynamic, routing-based depth optimization is a compelling direction for future work.

- **Cross-Architecture Generalization:** While we validated results on transformer models of different scales, the interaction between aggressive layer dropout and alternative architectures, such as Mixture-of-Experts (MoE) or non-transformer models, has not yet been explored.

- **Scaling Laws for Maximum Dropout:** We have not yet developed comprehensive scaling laws to predict the maximum dropout rate ($p_{max}$) that can be applied without incurring accuracy degradation relative to the dense baseline.

- **Scaling Analysis for Inference Benefits:** We have not quantified the different inference benefits (early exit loss, skipping intermediate layer loss, early exit adapter loss, and self-speculative decoding speedup) as TPP increases.

