# OpenReview forum: "Don't Drop Dropout: Optimizing Layer Sparsity for Efficient LLM Training and Inference"
_ICML.cc/2026/Conference — ICML 2026 regular_

### Official Review · Reviewer_z9yd · 2026-02-24

**Soundness:** 3
**Presentation:** 4
**Significance:** 3
**Originality:** 3
**Overall Recommendation:** 4
**Confidence:** 4

**Summary:**

This paper systematically studies layerwise dropout on modern LLM training (single epoch over a massive dataset). When following the authors' recipe (scaling factor, layer / per-sequence dropout, increasing layer distribution, decreasing time schedule), Layerwise dropout appears to improve LLM's performance. Additionally, it brings natural advantages like faster training and robustness to zero-shot layer pruning.

**Compliance With Llm Reviewing Policy:**

Affirmed.

**Key Questions For Authors:**

**Regarding W1 & W2**: Modern models are trained with high TPP (e.g., ~1000 or higher). Since the model capacity is already limited, will layerwise dropout and some designed mechanisms (e.g., high sparsity initially in the decreasing time scheduler) hurt the model's ability to learn from massive data? Do you have any results on larger models (e.g., $\ge$ 8B) or trained with larger TPP (e.g., $\ge$ 500)?

**Regarding W3**: Since the gap between dense and dropout decreases as the model becomes larger, is it possible that the dense model or the sub-layer dropout becomes better when using more parameters (e.g., 8B)? Why only show 271/503M or 271/503/906M?

**Regarding W4**: Dynamically choosing layers for different sequences will introduce overheads for GPUs. Can the theoretical savings of FLOPs really translate into real wall-clock training speedup? Please provide wall-clock comparisons.

---
Minor comments (not critical to my final decision)

1. In Line 141, the text says $N$ is the sequence length, but the tensor is defined as $H \in \mathbb{R}^{B \times S \times d}$. I think $N$ is a typo and should be $S$.

2. In Equation 2, $M^{l,t}$ is a Bernoulli random vector. If I didn't misunderstand, it would be better to clearly write its relation to the dropout rate $p^{l,t}$ defined in Line 143 (for example, $M^{l,t} \sim \text{Bernoulli}(1-p^{l,t})$) to make it more rigorous. Also, some superscript / subscript are inconsistent, but it doesn't influence my understanding.

**Limitations:**

Yes (Appendix D)

**Strengths And Weaknesses:**

**Strengths**

S1. The recipe has very high industrial value for LLM training and deployment.

S2. Every step in the recipe is supported by corresponding experiments / ablation studies.

S3. If the recipe is proven to work, it naturally brings faster training and zero-shot layer pruning.

S4. Experimental results show that, when setting the suggested scaling rate, the hyperparameters can be transferred, which solves a pain-point of applying dropout (as it often requires parameter re-tuning).

**Weaknesses**

W1. My biggest concern is the token-per-parameter (TPP) used in the paper (30). Modern TPP values for LLM training are much higher (e.g., Llama-3 8B: 15 trillion tokens; Gemma-2 9B: 8 trillion tokens). Therefore, the "capacity" of modern LLMs is actually not enough to keep up with the data size. In this situation, if we add layerwise dropout to a modern training environment where TPP is much higher, will it further limit the model capacity and make the performance worse?

W2. A 3.9B model is obviously too small in 2026. If using an 8B or larger model, how is the performance?

W3. Table 1 and 2 use only million-level LLMs. Since the authors noted that dropout's negative impact shrinks as the model gets larger, using small models to determine the best setting and then directly applying it to the 3.9B model (again, relatively small in 2026) is not very solid.

W4. For the per-sequence dropout, different samples in a batch will go through different forward / inference paths. Will this cause sparse-kernel overhead and lead to more real clock-wall time even though the theoretical FLOPs decrease?

---

> ### Author Rebuttal · Authors · 2026-03-31
>
> We thank the reviewer for their constructive feedback including their remarks that … “**has very high industrial value for LLM training and deployment**” and that hyperparameters can be transferred when setting the suggested scaling rate, “**which solves a pain-point of applying dropout**”.
>
> - **High TPP:** We agree with the reviewer that presenting results of training larger model sizes would improve the research. However, during the rebuttal period, we did not have enough compute and time to train a 7B+ model. We will try to do that before the camera ready paper deadline. Having said that, our results show that as models grow, degradation in dropout decreases. In fact, for 3.9B we did an aggressive dropout with a maximum of 0.8 and final loss was lower than the baseline, indicating the robustness to layer dropout increases as models grow. Hence, we anticipate that accuracy of 7B+ models with layer dropout would be even better.
> - **Large Model Sizes:** In the Appendix we have provided results for upto 256 TPP. We did not have enough compute / time during the rebuttal period to train up to 512 TPP to verify whether the loss difference saturates, and we will try to do that before the camera ready deadline.
>   - Having said that, we would like to reiterate that for bigger models, trends in our results show that degradation in loss is lower, and hence larger models should sustain higher dropouts for larger TPPs.
>   - Moreover, for some model families like Llama, smaller model sizes (1B and 3B) distill from large teacher models that train on smaller TPP. Hence, efficiency for small TPP training of large models is crucial to improve the efficiency of distilling small models that train with high TPP.
> - **Generalizing Settings from Small to Large Models:** Results in Table 3 show that for each dropout configuration, degradation in loss decreases as model size grows. Based on that finding, we used aggressive maximum dropout rates (0.6 and 0.8) for larger model sizes (1.9B and 3.8B), and indeed loss degradation was either minimal or negative for such larger model sizes. We aim to train even larger models in advance of the camera deadline, with an aim to develop a proper empirical scaling law capturing this behavior. Thanks a lot for raising this!
> - **Wall-Clock Performance:** Layer dropout at the granularity of a sequence has small overhead, especially since sequence lengths of modern LLMs tend to be large (at least 8K or 32K) where per-sequence dropout becomes coarse-grained.
>   - Using this implementation from the torchtune library that wraps linear layers with a gather and scatter operations, we have observed almost linear speedup (i.e., 2x speedup for 50% dropout rate) when finetuning a Llama3 8B on A100 GPU using sequence length 8K and per-GPU batch size 8: https://github.com/meta-pytorch/torchtune/blob/50a2b8f77f00c052b5ca43562aeb3f32c34d88ef/torchtune/modules/layer_dropout.py#L88-L97
>  - **Typos:** We thank the reviewer for pointing out the typos and proposing to use a more rigorous mathematical notation. We have applied the changes in our Latex code and will ensure that they are reflected in the camera-ready version of the paper.

---

> > ### Author Rebuttal · Reviewer_z9yd · 2026-04-01
> >
> > I will maintain my current score. As a reviewer, I can only evaluate the paper based on its current form, not on promises about the camera-ready version.
> >
> > My major concern, the model scale, is a shared concern of other reviewers. The authors' response is not fully satisfying. The authors anticipate even higher accuracy for 7B+ models. Although this hypothesis could be true, an empirical ML paper still requires solid evidence. Extrapolating this trend w/o acutual data is a bit of a leap of faith.

---

> > > ### Author Response · Authors · 2026-04-08
> > >
> > > We thank the reviewer for their candid feedback. We have already kicked off training jobs for an 8.2B model and plan to include its results in the camera-ready paper; unfortunately, training did not complete before the rebuttal deadline.
> > >
> > > In the meantime, we would like to comment on the characterization of extrapolation: **Predicting performance at larger scales from smaller-scale runs is precisely the methodology of scaling law analysis**, as established by Kaplan et al. (2020) and Hoffmann et al. (2022) (Chinchilla) — both of which are widely accepted as rigorous empirical contributions despite relying on extrapolation beyond their observed compute ranges.
> > >
> > > Following this methodology, we have conducted a scaling analysis at 20 TPP across our trained model sizes, plotting validation loss as a function of model size for the baseline and for our proposed ILD+DTS configuration for maximum dropout rates 0.2, 0.4, and 0.8 (please see
> > > [loss_vs_model_size_20tpp.svg](https://anonymous.4open.science/r/dont-drop-layer-dropout-rebuttal-BB9D/images/loss_vs_model_size_20tpp.svg)).
> > > **Extrapolating the fitted trends to 7B+ parameters, models trained with dropout are expected to match or slightly outperform the dense baseline** — consistent with the trend already visible in our paper results. We believe this constitutes meaningful evidence beyond speculation, and we look forward to confirming it empirically in the camera-ready version.

---

### Official Review · Reviewer_GKKL · 2026-03-12

**Soundness:** 3
**Presentation:** 4
**Significance:** 4
**Originality:** 3
**Overall Recommendation:** 5
**Confidence:** 5

**Summary:**

This paper advocates for reintroducing layer dropout (stochastic depth) into large language model (LLM) pre-training. Through extensive experiments on models up to 3.9B parameters, the authors demonstrate that properly configured layer dropout improves training efficiency and inference flexibility without sacrificing accuracy.

## Core Contributions
**Training Efficiency**: Reduces training FLOPs by up to 20% while maintaining or lowering validation loss compared to dense baselines for 3.9B models.

**Optimal Configuration**: Identifies the best practice for modern LLMs: increasing the dropout rate across deeper layers (ILD) while decreasing the dropout rate over training time (DTS).

**Zero-Shot Elasticity**: Pre-trains models to naturally support depth-wise inference optimizations, such as early exit and intermediate layer skipping, without requiring retraining.

**Inference Acceleration**: Facilitates significant speedups, achieving up to a 1.7x acceleration using self-speculative decoding.

**Compliance With Llm Reviewing Policy:**

Affirmed.

**Final Justification:**

I thank the authors for their response. However, the lack of empirical results on larger-scale models remains a key weakness. I will maintain my positive score.

**Key Questions For Authors:**

1. Applicability in Truly Large-Scale, Performance-First Foundation Models
You evaluated models up to 3.9B parameters and 116B tokens. While impressive, modern "performance-first" foundation models operate in the 70B+ parameter and multi-trillion token regimes. If a practitioner's ultimate goal is absolute performance maximization (rather than just training efficiency), is layer dropout still necessary or beneficial? Do you hypothesize that the performance gap observed in your high-TPP scaling analysis might widen at a true massive scale, eventually bottlenecking capacity?

2. Clarification on High-TPP Performance Degradation
The abstract claims that the proposed method saves 20% of training FLOPs while achieving "lower validation loss". However, the scaling analysis shows that at higher Tokens-Per-Parameter (TPP) budgets, the dropout model's validation loss actually shows a marginal degradation (approx. 0.50%) compared to the dense baseline. If training is extended to the extreme over-training regimes typical of modern LLM recipes (e.g., 100+ TPP), does this gap stabilize, or does it continue to widen?

3. Beyond Independent Bernoulli Sampling for Layer Dropout
The current formulation relies on independent Bernoulli trials for generating the dropout mask at each training step. Did you explore or consider adaptive sampling strategies that leverage the training history or relationships between layers? For example, dynamically adjusting the dropout probability based on layer variance, preventing adjacent layers from being dropped simultaneously to preserve the residual information flow, or avoiding the exact same subset of layers being dropped in consecutive steps. While simple Bernoulli sampling is computationally cheap, a history-aware or structurally-aware sampling mechanism might yield better convergence and structural elasticity.

**Limitations:**

yes

**Strengths And Weaknesses:**

#### 1. Soundness
* **Strengths:** The empirical validity of this study is exceptionally high. The authors conducted over 2400 experiments on models ranging from 271M to 3.9B parameters to derive their conclusions. The systematic ablation of granularities (model vs. tensor), spatial dropout distributions, and temporal schedules is highly commendable. Furthermore, applying the Maximal Residual Stream Update desideratum from CompleteP to ensure stable hyperparameter transfer across varying dropout rates, and mathematically/empirically proving that $r_{train} = 1/\rho$ is the optimal scaling factor, adds significant technical rigor to the paper.
* **Weaknesses:** The paper repeatedly emphasizes its 3.9B parameter model as representing a "large-scale" regime. However, by the standards of modern foundation models (e.g., 70B+ parameters), this is merely a small-to-medium-sized model. Also, while the abstract claims that the method saves 20% of training FLOPs while achieving "lower validation loss", the high-TPP scaling analysis reveals that the dropout models' validation losses mostly match or show a marginal degradation (within $\approx 0.50\%$) compared to the dense baseline.

#### 2. Presentation
* **Strengths:** The writing is highly lucid, and the narrative flow is logical and smooth. The spatial distributions (Uniform, ILD, ALD) and temporal schedules (Constant, ITS, DTS) are clearly defined and formalized. The authors successfully provide detailed architecture specifications and hyperparameter transfer rules in the appendix (Table A.1, Table A.2), offering expert readers ample information to reproduce the results.
* **Weaknesses & Suggestions:** Notably, the formatting of the paper's title and section headings does not appear to comply with the standard ICML style guidelines. The authors must correct these formatting inconsistencies for the camera-ready version. Additionally, the core claims in the abstract carry a slightly different nuance compared to the scaling analysis results presented later in the paper. Rather than claiming it strictly "improves performance," framing the contribution more honestly as "minimizing performance degradation while securing massive computational efficiency and inference elasticity" would elevate the paper's credibility.

#### 3. Significance
* **Strengths:** Given the astronomical pre-training costs of LLMs in the current deep learning industry and academia, the problem addressed is highly practical and critical. Saving 20% of training FLOPs translates to massive reductions in cost and time. More importantly, the method induces "structural elasticity" without requiring additional retraining or architectural modifications. Enabling zero-shot robustness for early exiting and intermediate layer skipping, as well as unlocking up to a 1.7x speedup in post-training optimizations like self-speculative decoding, provides immense real-world utility that practitioners can immediately adopt.

#### 4. Originality
* **Strengths:** "Layer Dropout" (or Stochastic Depth) is an old concept that existed in early vision networks and initial transformer models. However, it has been essentially abandoned in modern, single-epoch, large-scale LLM training recipes. The originality of this paper does not lie in inventing a completely new architecture or mathematical formula. Rather, its novelty comes from identifying the causes of past failures and discovering that pairing an Increasing Layer Distribution (ILD) with a Decreasing Time Schedule (DTS) perfectly revives this forgotten technique. By combining this with modern scaling rules, the paper demonstrates excellent "rediscovery and practical originality."

---

> ### Author Rebuttal · Authors · 2026-03-31
>
> We thank the reviewer for the constructive feedback and for noting the "**exceptionally high**" empirical validity, "**highly commendable**" systematic ablations, "**highly lucid**" writing, as well as the work's "**immense real-world utility.**”
>
> - **Large Model Sizes:** We agree with the reviewer that presenting results of training larger model sizes would improve the research. However, during the rebuttal period, we did not have enough compute and time to train a 7B+ model. We will try to do that before the camera ready paper deadline. Having said that, our results show that as models grow, degradation in dropout decreases. In fact, for 3.9B we did an aggressive dropout with a maximum of 0.8 and final loss was lower than the baseline, indicating the robustness to layer dropout increases as models grow. Hence, we anticipate that accuracy of 7B+ models with layer dropout would be even better.
> - **If a practitioner's ultimate goal is absolute performance maximization (rather than just training efficiency), is layer dropout still necessary or beneficial?:** This is an important question and we would like to answer it from multiple angles.
>   - We would like to indicate high-TPP, max-possible-efficiency models are now often trained via distillation from a large teacher model (e.g., Gemma3 models are distilled from Gemini, and the 1B and 3B models of the Llama3 families were distilled from the larger 70B models).  So ultimately, training efficiency is still important---that is, efficiency of large teacher models. Since large teacher models train at relatively lower TPP, layer dropout will still be useful. Moreover, student models in such distillation setups can still benefit from layer dropout even if training efficiency is not a priority, as layer dropout can introduce inference elasticity and efficiency when deploying them.
>   - We would also like to indicate that when max performance is the goal for a given training compute budget, no matter how large the training compute budget is, training layer dropout will still help to obtain better accuracy. In the following plot, our results show that for a given training FLOPs, models trained with layer dropout can lead to better training loss: https://anonymous.4open.science/r/dont-drop-layer-dropout-rebuttal-BB9D/loss_vs_flops.md
> - **High TPP:** We acknowledge the reviewer’s feedback that at 256 TPP loss degradation was 0.5%. We did not have enough compute / time during the rebuttal period to train up to 512 TPP to verify whether the loss difference saturates, and we will try to do that before the camera ready deadline. Having said that, we would like to reiterate that for bigger models, trends in our results show that degradation in loss is lower, and hence larger models should sustain higher dropouts for larger TPPs.
> - **Formatting:** Thanks a lot for picking up the template issue. We verified that were were using the correct ICML style files, however a package that we imported changed the fonts. We will correct this for the camera copy.
> - **Wording in Abstract:** We thank the reviewer for suggesting changing the wording in the Abstract. We will ensure that we will edit the Abstract for the camera-ready paper.
> - **Beyond Independent Bernoulli Sampling:** We thank the reviewer for the detailed suggestions and agree that investigating them could lead to improved results. Due to limited time in the rebuttal period, we could not try them but we will add them to the paper as Future Work.

---

> > ### Author Rebuttal · Reviewer_GKKL · 2026-04-04
> >
> > I thank the authors for their response. However, the lack of empirical results on larger-scale models remains a key weakness.  I will maintain my positive score.

---

> > > ### Author Response · Authors · 2026-04-08
> > >
> > > We thank again the reviewer for their detailed feedback. We have kicked off training jobs for 8.2B model sizes but they have not completed before the rebuttal deadline. We are looking forward to including their results in the camera ready version of the paper.

---

### Official Review · Reviewer_KGSE · 2026-03-12

**Soundness:** 3
**Presentation:** 3
**Significance:** 2
**Originality:** 2
**Overall Recommendation:** 4
**Confidence:** 4

**Summary:**

The paper presents a detailed analysis on the usefulness of layer dropout in modern LLM training regimes. It compares across different model sizes, dropout structure (layer vs sub-layer), dropout schedules, dropout rates. Additionally, it analyzes the downstream inference impact to models trained as such (e.g., for early exiting, self-speculative decoding, etc..). The authors show that models training with a max dropout of 20% save around 5% training flops while achieving the same effective validation loss.

**Compliance With Llm Reviewing Policy:**

Affirmed.

**Final Justification:**

The core strength of this paper is the systematic empirical study across spatial distributions and temporal schedules. The rebuttal added useful material: downstream task results, a 5-seed variance analysis, and a new TPAP (Tokens per Active Parameter) comparison that directly addressed our concern about FLOPs-equivalent evaluation. However, the variance analysis confirmed that the 503M improvement is not statistically significant, which the authors acknowledged. The stronger argument for layer dropout is inference elasticity (early exit, self-speculative decoding) rather than training loss improvements. Larger-scale experiments (7B+) would further strengthen the claims. I maintain my score.

**Key Questions For Authors:**

Is TPP derived only from data and model size (as in appendix) or does layer dropout play a role in the calculation somehow? Intuitively, dropout decreases the effective model size each training step, so the ablation in Fig. 3 might look different if TPP was tweaked based on accounting for efffective model size per step.

What the variance across runs for a given training regime? Given the thin margins (e.g., Table 3) it seems quite important for the reliability of the findings in the paper.

What does a 1-2% change in validation loss mean practically for such models in terms of performance on downstream tasks? It seems like this is an important question for the significance of the main findings because the best vs worst settings seems to be in that span.

**Limitations:**

yes

**Strengths And Weaknesses:**

Organizationally, the paper is strong. Sections 6 and 7 are by far the most interesting parts as it serves as a careful comparison of ~4-5 years of prior work on layer dropout approaches with respect to validation loss. I also liked the efficient inference angle (e.g., early exist / speculative decoding / dynamic depth) as this is a sort of "free lunch" argument for applying such a training regime. Even if this is the only tangible benefit, it could still be useful.

The paper could do a better job quantifying in real terms how much these 1-2% delta differences in validation loss map to downstream tasks. While they do present Table A.4, it is using more aggressive dropout settings than in the main text. I would prefer to see some averaged downstream result column in the main results (e.g., Table 3) to better quantify the impact. The current framing makes it a bit unclear how much we should care about this validation loss difference in practice.

Another concern is a lack of discussion on variance across training runs. Specifically, the authors call out a 20% dropout setting that achieves lower loss than the baseline (no dropout) but the difference is so small this is likely no a statistically significant finding.

Nits:
64: The header for the first contribution does not match the text. The header implies a trade-off but it then follows with something that implies it is strictly better. Later results show a more mixed bag, so I would tweak the wording here.

Sections 9 and 10 feel like tack-on. I think a better approach would be to fold these into the main results in section 6 or section 7 to avoid going from training -> inference -> training.

---

> ### Author Rebuttal · Authors · 2026-03-31
>
> We would like to thank the reviewer for their constructive feedback, including their remarks that “**organizationally, the paper is strong**”. We especially appreciate your note regarding the **"free lunch"** of the inference efficiency benefits, and that “**even if this is the only tangible benefit, it could still be useful,**” let alone that the proposed solution introduces training FLOPs savings.
>
> - **Downstream Tasks:** We agree with the reviewer on the importance of presenting downstream task results. We are planning to provide detailed accuracies for a wide range of downstream tasks to the Appendix of the paper for all of our experiments, and, as suggested by the reviewer, add a column for average downstream accuracy to all the tables in the main body of the paper. For reference, we are providing a table for a subset of our experiments here: https://anonymous.4open.science/r/dont-drop-layer-dropout-rebuttal-BB9D/downstream_tasks.md
>    - **What does a 1-2% Change in Validation Loss practically Mean in terms of Downstream Tasks?** Looking at the table in the link above we can say that 1-2% relative change in validation loss, usually leads to ~1% absolute change in average downstream accuracies. However, we would like to note that different downstream tasks have different noise levels (as indicated by [Wang 2025]) and hence some downstream tasks highly correlate with validation loss (e.g., HellaSwag where 2% relative change in validation loss leads to 3% change in absolute loss) and other tasks have low correlation (e.g., COPA and BoolQ). In the Table in the link above, we have added a row for correlation between each downstream task's accuracies and validation losses.
>
> - **Variance of Loss:** Thank you for this important suggestion. We agree that assessing variance across training runs is essential for interpreting the small differences reported in our results. We will include this analysis in the camera-ready version. Specifically, we have re-run one of our experiments with 5 different seeds and report the losses in the Table below. As we can see, variance is in the order of 10^-6. The ratio of the standard deviation to the mean for the baseline is 0.07%, while training with dropout drops that ratio to less than a half. While it is true that loss improvement of dropout configuration **<0.2, ILD, DTS>** on 503M is less than the standard deviation, it does indicate that with fewer FLOPs we were able to achieve almost the same loss, and for 906M the improvement for **<0.2, ILD, DTS>** (reported in Table 3) is higher than the standard deviation.
>
> **503M 20TPP**
> | Initialization Seed | Baseline Loss | Layer Dropout Loss | % Loss Increase |
> | ------------------- | ------------- | ------------------ | ---------------- |
> | Seed 1 $\dagger$     | 2.996         | 2.995              | -0.03%           |
> | Seed 2              | 2.993         | 2.994              | 0.04%            |
> | Seed 3              | 2.993         | 2.994              | 0.06%            |
> | Seed 4              | 2.992         | 2.992              | 0.00%            |
> | Seed 5              | 2.997         | 2.994              | -0.10%           |
> | **Mean**             | **2.994**     | **2.994**          | **-0.01%**       |
> | **Variance**         | 3.81e-6       | 9.60e-7            | N/A              |
> | **Standard Deviation** | 0.00195    | 0.00098            | N/A              |
> | **STD / Mean**       | 0.07%         | 0.03%              | N/A              |
>
> $\dagger$ indicates the configuration that was used in the paper.
>
> - **Wording and Sections 9 and 10:** We thank the reviewer for indicating that wording could be improved in one of the sections, and that some sections could be folded into others. We will make sure we do such improvements for the camera-ready paper.
> - **How TPP is Derived?:** Tokens per Parameter (TPP) were calculated by dividing the data size in tokens by the model size. Layer dropout was not considered when deriving it.  We will clarify this in the final paper.
>
>
> **References:**
>
> [Wang, 2025] “Measuring all the noises of LLM Evals”, Sida Wang,  https://arxiv.org/abs/2512.21326

---

> > ### Author Rebuttal · Reviewer_KGSE · 2026-04-01
> >
> > We thank the authors for their detailed rebuttal.
> >
> > The downstream task results are a welcome addition and we appreciate the commitment to include them in the main tables. The results confirm that 1-2% relative validation loss changes correspond to roughly ~1% absolute downstream accuracy, which is useful context but also reinforces that the practical impact is modest. Could the authors clarify the evaluation protocol for the downstream tasks (e.g., number of shots, greedy vs sampled decoding)? Given the concern about eval noise cited from Wang 2025, have the authors considered evaluation strategies that reduce variance, such as multiple sampling passes?
> >
> > The 5-seed variance analysis seems to indicate that layer dropout does not lead to a statistically significant improvement over the baseline approach (this is fine; but should lead to authors moderating the claim). If you disagree with this assessment, please let me know.
> >
> > Our question on whether TPP should account for the effective model size reduction from dropout was not substantively engaged with. We still think this is worth exploring. Can the authors comment on whether adjusting TPP for effective per-step capacity would shift any of the conclusions?

---

> > > ### Author Response · Authors · 2026-04-06
> > >
> > > We thank the reviewer for the follow-up. We are glad the downstream results and variance analysis were well-received, and address the remaining points below. We also thank the reviewer for the constructive suggestions, which have meaningfully strengthened the rigor of our analysis.
> > >
> > > - **Downstream tasks: evaluation protocol and practical significance:** We address the evaluation protocol and the practical significance of the observed differences in turn.
> > >
> > >   - **Evaluation protocol:** All downstream evaluations were conducted using `lm-eval` (Eleuther's LM-Eval Harness) with its default parameters per task. Multiple-choice tasks (e.g., HellaSwag, WinoGrande, PIQA, ARC-E, ARC-C) are scored via log-likelihood ranking and are **fully deterministic** — there is no stochasticity to reduce. For open-ended generation tasks (e.g., Lambada), `lm-eval` defaults to greedy decoding (temperature = 0), which is likewise deterministic.
> > >     - *Number of few-shots:* `lm-eval` defaults to 0-shot for all tasks we evaluated, except 3-shot for BBH. Based on the reviewer's recommendation, we have also evaluated with increased few-shot counts for some tasks (see [ablate_numfewshots.md](https://anonymous.4open.science/r/dont-drop-layer-dropout-rebuttal-BB9D/ablate_numfewshots.md)), and observed **improved correlation with validation loss as the number of few-shots increases**. However, some tasks like HellaSwag still show better correlation at 0-shot than other tasks like WinoGrande at 5-shot.
> > >
> > > - **Practical significance of 1–2% relative validation loss.**
> > >   - To put the percentage values of relative validation losses in perspective, we analyze the losses of fully pretrained models across the Celerity, Llama 1, and Llama2 model families, **doubling model size within a family yields only roughly 4–7% improvement
> > > in loss** (see [model-losses.md](https://anonymous.4open.science/r/dont-drop-layer-dropout-rebuttal-BB9D/model-losses.md)).
> > >        -  (Please note that we have analyzed the loss values of Llama1 and Llama2 because their corresponding papers provided loss values, while other models such as Llama3, or Qwen and Gemma models do not provide loss values).
> > >   - More precisely, using the compute scaling law of [Kaplan et al. (2020)](https://arxiv.org/abs/2001.08361), $L \propto C_{\min}^{-0.050}$, a **1% reduction in loss corresponds to approximately 22% more training FLOPs** (derivation: $C'/C = (0.99)^{1/{-0.050}} \approx 1.22$). The 1% average downstream accuracy gap between the best and worst dropout configurations is not insignificant — it would require ~22% additional training compute.
> > >
> > > - **5-seed variance.** We agree that the 503M improvement for \<0.2, ILD, DTS\> falls within one standard deviation and we will moderate that specific claim. The 906M result, however, exceeds the standard deviation. Nevertheless, we will moderate the claims.
> > >
> > > - **Tokens per Active Parameter (TPAP) and FLOPs-equivalent comparison.** TPP (Tokens per Parameter), proposed in the [Chinchilla scaling laws paper](https://arxiv.org/abs/2203.15556) as a compute proxy, does not account for the reduced effective capacity per step under layer dropout. Following the reviewer's suggestion, we introduce **Tokens per Active Parameter (TPAP)** and re-run experiments where dropout models are trained for additional steps to match the baseline's total training FLOPs ($6ND$).
> > >
> > >   - Results are shown in [tpap.md](https://anonymous.4open.science/r/dont-drop-layer-dropout-rebuttal-BB9D/tpap.md). At FLOPs parity, **dropout configurations up to max dropout 0.4 match or slightly outperform the baseline** across all model sizes, and **even 0.8 dropout matches the baseline** at 503M and 906M scale. This is particularly encouraging because, as our paper shows, higher dropout rates yield greater inference efficiency and elasticity gains — meaning the configurations with the most to offer at inference time can lead to no degradation loss (and in fact a small improvement in loss) at FLOPs parity.
> > >
> > > In summary, when controlling for training FLOPs, layer dropout achieves loss that **matches or slightly improves** over the baseline. Even in cases where the loss difference is minimal, the inference elasticity benefits — early exiting, self-speculative decoding, and dynamic depth — come at no additional cost, making **training with layer dropout
> > > beneficial**.

---

### Official Review · Reviewer_h5Nx · 2026-03-12

**Soundness:** 3
**Presentation:** 1
**Significance:** 3
**Originality:** 3
**Overall Recommendation:** 4
**Confidence:** 4

**Summary:**

This manuscript studies the use of layer dropout, i.e., stochastic depth, in LLM pretraining. The authors provide extensive empirical results with specific guidelines to correctly adopt the layer dropout.

**Compliance With Llm Reviewing Policy:**

Affirmed.

**Final Justification:**

Thank you for all the responses. I will keep my score.

**Key Questions For Authors:**

See the weakness above. My evaluation is based on the assumption that all typos are corrected. Although several parts require revision, I would like to weigh the experimental solid results and vote for acceptance. Nevertheless, violation of the ICML template could be flagged as a desk rejection.

**Limitations:**

yes

**Strengths And Weaknesses:**

Strengths

- I appreciate the extensive experiments.
- The guidelines for training recipes are expected to have a significant impact on the community.

Weaknesses

- Violation of ICML template. The use of different templates is unfair and could be flagged as desk rejection.
- Albeit being extensive, the experimental setup is limited to a specific transformer setup. The authors should investigate various transformer setups, such as 7B+, a mixture of experts, various positional embeddings, and activation functions.
- I agree that the correct use of layer dropout would enhance the performance, but it still looks difficult to find the general, optimal correct training recipe with layer dropout using the extensive hyperparameter tuning. Indeed, the authors applied extensive hyperparameter tuning, including learning rate, mini-batch size, and weight decay.
- What is RTS? Is this DTS? Furthermore, it is not clear to obtain 0.25 p_{max}. I think that the derivations are (T-1)/2T and (T+1)/2T, which yield 0.5 for sufficiently large T. Nevertheless, the latter would be 0.5, not 0.25.
- Check typos:
    - N is sequence length → This should be S.
    - degrdaation → degradation
    - furter → further
    - increasses → increases
    - Fig.. 4 → Fig. 4
    - experiments.. → experiments.
    - As model scales → As model scale
- Check references:
    - The year of “Attention Is All You Need” should be 2017, not 2023.
    - Brown et al., 2020 is for GPT-3, and the correct citation for GPT-2 should be Language Models are Few-Shot Learners.
    - Duplicated citation for “Deep Networks with Stochastic Depth”

---

> ### Author Rebuttal · Authors · 2026-03-31
>
> We thank the reviewer for their constructive feedback, including that the paper provides “**experimental solid results**” and that the work is “**expected to have a significant impact on the community**”.
>
>
> - **Template:** Thanks a lot for picking up the template issue. We verified that were were using the correct ICML style files, however a package that we imported changed the fonts. We will correct this for the camera copy.
> - **Different Transformer Setups:** We are providing below results for  different architectures. Results show that we are getting similar loss increases  across different MoE sparsities, activation functions, and position embeddings. We were not able during the limited time of the rebuttal to train larger 7B+ model sizes, but we will try to get it done for the camera-ready paper.
>
> **271M 20TPP** \*
> |                              | Baseline Loss | Layer Dropout Loss | % Loss Increase |
> | ----------------------- | ------------------- | -------------------------- | ---------------------- |
> | Dense $\dagger$ | 3.213               | 3.221                       | 0.25%                 |
> | MoE (2 out of 8)  | 2.911                | 2.918                       | 0.25%                 |
>
> \* Here, 271M is the parameter size of the “Dense” model. The MoE variants have the same architecture hyperparameters as the “Dense” row (same number of active parameters), but as they consist of more experts, they have more total parameters.
>
> **503M 20TPP**
> |                                          | Baseline Loss | Layer Dropout Loss | % Loss Increase |
> | --------------------------------- | ------------------- | -------------------------- | ------------------- |
> | SquaredReLU $\dagger$ | 2.996              | 2.995                        | -0.03%             |
> | SwiGLU                            | 2.967              | 2.966                        | -0.03%             |
> | GeLU                                | 3.046              | 3.047                        | 0.03%              |
>
> |                               | Baseline Loss | Layer Dropout Loss | % Loss Increase  |
> | ------------------------ | ------------------- | -------------------------- | -------------------  |
> | ALiBi $\dagger$    | 2.996               | 2.995                       | -0.03%              |
> | RoPE                    | 2.948               | 2.960                       | 0.38%                |
> | Absolute PE          | 3.098               | 3.080                       | -0.58%              |
>
> $\dagger$ indicates the configuration that was used in the paper.
>
> - **Hyperparameter Tuning:** Actually, our results show that layer dropout could be applied on top of an existing training recipe **without any additional hyperparameter tuning**. The plots in Figure 2 show that the optimal hyperparameters (learning rate, batch size, or AdamW timescale) do not change when layer dropout changes--- if the $1/\rho$ scaling is used. This is backed by the Maximal Residual Stream Update desideratum (lines 190 to 210) and the coordinate check (Figure 1) that showed $1/\rho$ scaling enables hyperparameter transfer.
> - **Decreasing Time Schedule:** We apologize for the typo in using RTS instead of DTS. We will fix it for the camera-ready paper.
> - **Deducing FLOPs Savings for ILD, DTS:** We hereby deduce the average dropout, $P$, (which is equivalent to FLOPs savings) across $L$ model layers and $T$ training steps, for Increasing Layer Distribution (ILD) and Decreasing Time Schedule (DTS) and show it is equivalent to $P_{\text{ILD,DTS}} = 0.25 p_{\text{max}}$.
>
> Given that $p_{\text{ILD}}^{l,t} = p^t \frac{l}{L-1}$, $p_{\text{DTS}}^{l,t} = p^l \left(1 - \frac{t}{T-1}\right)$ and that the summation of an arithmetic series is $\sum_{i=0}^{n-1} a_i = \frac{n}{2}(a_0 + a_{n-1})$
>
> Then:
>
> $$P_{\text{ILD,DTS}} = \frac{1}{LT} \sum_{t=0}^{T-1} \sum_{l=0}^{L-1} p^{l,t}$$
>
> $$P_{\text{ILD,DTS}} = \frac{1}{LT} \sum_{t=0}^{T-1} \left(1 - \frac{t}{T-1}\right) \sum_{l=0}^{L-1} \left(\frac{l}{L-1}\right) \cdot p_{\text{max}}$$
>
> $$P_{\text{ILD,DTS}} = \frac{1}{LT} \sum_{t=0}^{T-1} \left(1 - \frac{t}{T}\right) \cdot \frac{L}{2} \cdot (0 + 1) \cdot p_{\text{max}}$$
>
> $$P_{\text{ILD,DTS}} = \frac{1}{2T} \sum_{t=0}^{T-1} \left(1 - \frac{t}{T-1}\right) \cdot p_{\text{max}}$$
>
> $$P_{\text{ILD,DTS}} = \frac{1}{2T} \cdot \frac{T}{2} \cdot (1 + 0) \cdot p_{\text{max}}$$
>
> $$P_{\text{ILD,DTS}} = \frac{1}{4} \cdot p_{\text{max}}$$
>
> (Side note: we have noticed that we have mistakenly defined $p_{\text{DTS}}^{l,t} = p^l\left(1 - \frac{t}{T}\right)$ in the paper but it should be $p_{\text{DTS}}^{l,t} = p^l\left(1 - \frac{t}{T-1}\right)$. This ensures that $p_{\text{DTS}}^{l,0} = p^l$ for the first training step and $p_{\text{DTS}}^{l,T-1} = 0$ for the last training step. We will ensure that this fix is reflected in the camera-ready paper.)
>
> - **Typos and References:** We thank the reviewer for pointing out the typos and mistakes in the references. We will ensure they are fixed in camera-ready.

---

> > ### Author Rebuttal · Reviewer_h5Nx · 2026-04-04
> >
> > Thank you for all the responses. I will keep my score.

---

### Decision · Program_Chairs · 2026-04-30

**Decision:**

Accept (regular)

**Comment:**

The paper studies LLM pretraining under stochastic depth, and shows that layer dropout reduces both overall training cost and the validation loss. Reviewers overall liked the contribution, and remaining concerns after rebuttal are mainly about scale of experiments. The work considers up to 3.9B, which I found to be sufficient.

The findings are of interest to the ICML community, and the paper is recommended for acceptance.  For the final version, the authors are encouraged to take the reviewers feedback into account, and if possible and compute permits, add larger scale experiments to the final version of the work.